# Lexical-semantic properties of verbs and nouns used in conversation by people with Alzheimer's disease

Eric Williams[1]*, Catherine Theys[1,2], Megan McAuliffe[1,2]

1 School of Psychology, Speech and Hearing, University of Canterbury, Christchurch, New Zealand, 2 New Zealand Institute of Language, Brain and Behaviour, University of Canterbury, Christchurch, New Zealand

* eric.williams@pg.canterbury.ac.nz

**Data Availability Statement:** We are unable to make our data publicly accessible, as the data originate from a third-party source and are not owned by us. Our data originate from the Carolina Conversations Collection (CCC), https://

## Abstract

Alzheimer's disease (AD) is accompanied by language impairments and communicative breakdowns. Research into language processing by people with AD (pwAD) has focused largely on production of nouns in isolation. However, impairments are consistently found in verb production at word and sentence levels, and comparatively little is known about word use by pwAD in conversation. This study investigated differences between pwAD and cognitively healthy controls in conversational use of nouns, verbs, and pronouns. Speech samples produced by 12 pwAD and 12 controls for the Carolinas Conversations Collection were analysed for noun, verb and pronoun counts and ratios, lexical diversity overall and among nouns and verbs, copula use, and frequencies and ages of acquisition (AoA) of nouns and verbs produced. pwAD used fewer nouns and a narrower range of words than controls, exhibiting signs of increased reliance on pronouns and decreased noun diversity. Age affected noun frequencies differently within each group—pwAD produced nouns of lower frequencies with age, while controls produced nouns of higher frequencies. pwAD produced nouns of higher AoA than controls. Verb use differed little by group. These findings highlight the need to account for differences between nouns and verbs, including in frequency, AoA, proportion of all words spoken, and context-dependent processing demands, when drawing conclusions on language use by pwAD. They also suggest potential for communicative interventions targeting contextual use of both nouns and verbs.

## Introduction

Alzheimer's disease (AD) often results in communicative breakdowns that negatively impact the person with AD (pwAD) and caregivers, including by discouraging interactions that could help to preserve dignity and maintain relationships [1–3]. These breakdowns appear at least partially attributable to declining amounts of informative content in speech produced by pwAD. In connected speech, pwAD have been found to produce fewer words than controls, with total output decreasing as the disease progresses [4, 5]. In the speech they do produce, pwAD describe events less accurately, producing fewer information units than controls and

carolinaconversations.musc.edu/ccc/about/. This is a digital archive of recorded interviews with American men and women aged 65 and older, many of whom are experiencing cognitive decline or other health issues. Our study involved analysis of speech samples from the CCC, and our data files include all words produced in the speech samples analysed. Even with that information removed from our data files, speech samples can be rebuilt wholly or in part using data relevant to the reported study. Researchers can gain access to the CCC by following the process described here: https://carolinaconversations.musc.edu/ccc/help/access/approval. Once researchers have been granted CCC access, reasonable requests for our study-specific data can be directed to any of the study's authors.

**Funding:** The author(s) received no specific funding for this work.

**Competing interests:** The authors have declared that no competing interests exist.

thus omitting relevant information [6, 7]. These declines in informative content have resulted in characterizations of spoken discourse by pwAD as vague or empty [6, 8].

Knowledge of how specific lexical-semantic changes affect the informative content of speech produced by pwAD may help improve diagnosis and monitoring of the disease [6, 9]. An improved understanding of the nature of communicative breakdowns can also facilitate interventions that improve communication between pwAD and their caregivers [10, 11]. Consensus findings from studies of single-word tasks suggest impaired production of both verbs and nouns by pwAD compared to controls [7]. Within groups, pwAD tend to be less accurate with verbs than nouns [7]. Impairments in discrete word production reflect effects of connectedness in the semantic network and contributing psycholinguistic properties [7, 12]. Word frequency and age of acquisition (AoA) are believed to influence semantic network development, and pwAD perform less accurately on naming tasks requiring retrieval of words that are less frequent or are acquired later in life [13]. Better performance with nouns has been attributed to stronger relationships within this word class; however, psycholinguistic effects are unclear, as verbs tend to be more frequent than nouns, but are acquired later in life [14].

Lexical-semantic changes are also present in discourse produced by pwAD. Consensus findings indicate that pwAD produce more verbs and fewer nouns than controls in both picture descriptions and spontaneous speech [9, 15–17]. Declines in noun production are accompanied by increased reliance on pronouns, which function as less specific noun substitutes [9, 16, 17]. Findings from picture descriptions suggest that pwAD also use a less diverse range of words than controls in discourse [9, 16]. These changes are accompanied by psycholinguistic differences in the groups' speech samples. In picture descriptions and story retellings, pwAD rely on simple, generic words including copulas and verbs and nouns of higher frequencies than those by controls [18, 19]. Interestingly, findings from picture description tasks have not indicated that pwAD rely on words of lower AoA [9, 20]. Fraser et al. [9] included AoA in their investigation of language features that might help distinguish pwAD from healthy older people but did not find it to contribute to identification of pwAD. Yeung et al. [20], meanwhile, actually found pwAD to use words of higher AoA than controls, a finding they attributed to a tendency by pwAD to produce utterances that were inappropriate to the task. However, consensus findings on word use have led to suggestions that breakdowns in semantic representations of "advanced," or less well-connected, verbs and nouns in AD lead to their replacement with easily accessible, more well-connected, but less specific alternatives [9, 18]. This process would result in frequent reuse of less advanced verbs and nouns and increased use of pronouns, suggesting that changes in POS reliance, lexical diversity, and psycholinguistic properties of words produced by pwAD are related [9, 16].

In everyday communication, this combination of changes would contribute to perceptions of speech by pwAD as uninformative, inaccurate, or vague. However, little information is available on lexical diversity or psycholinguistic properties of words produced by pwAD in spontaneous speech. Where reported, these findings have rarely been broken down by POS. Findings on lexical diversity include just one report of reduced overall diversity ($n$ = 8 pwAD) and one of reduced diversity among both nouns and verbs ($n$ = 10 pwAD) [15, 17]. Despite the findings of word frequency and AoA effects on performance in single-word tasks, these measures have not been considered in analyses of spontaneous speech, either overall or by POS [21]. Breakdowns of diversity, frequency, and AoA of nouns and verbs produced in spontaneous speech would reveal whether the differential effects seen in discrete word production manifest in communicative situations. This knowledge would be useful in the design of targeted interventions. POS breakdowns would also help determine the extent to which overall changes in diversity, frequency, and AoA are attributable to changes in POS quantities. This is of

interest especially considering the likely overuse by pwAD of a limited number of high-frequency, low-AoA pronouns.

The present study therefore aims to investigate whether pwAD differ from controls in the use of words of different POS in conversational speech. Five hypotheses are investigated.

- H1 predicts that pwAD will use significantly fewer nouns and significantly more verbs and pronouns than controls, with these changes resulting in significantly lower N/V ratios.

- H2 predicts that pwAD will exhibit significantly decreased lexical diversity compared to controls on three measures—overall, for nouns, and for verbs.

- H3 predicts that pwAD will produce significantly more copulas than controls.

- H4 predicts that pwAD will use nouns and verbs of significantly higher frequencies than controls.

- H5 predicts that pwAD will use nouns and verbs of significantly lower AoAs than controls.

## Methods

### Dataset

Language data used in this study come from the Carolinas Conversations Collection (CCC), a digital archive of recorded interviews with American men and women aged 65 and older [22, 23]. This can be found at https://carolinaconversations.musc.edu/ccc/about/. Data collection was approved by institutional review boards at the University of North Carolina at Charlotte and the Medical University of South Carolina. Participants who were able to write provided written informed consent to recording, as did guardians of all participants. Each participant also provided oral assent to recording.

The CCC includes interviews with participants diagnosed with dementia by a physician [23]. Prior to interviewing participants, researchers agreed not to conduct further cognitive testing; however, pwAD are uniformly described as being in moderate to late disease stages [23]. The CCC also includes interviews with participants who were screened to rule out dementia [22]. Conversations revolve around daily life and health. Interviewers disclose their own typical daily activities before asking interviewees to do the same. Where an interviewee refers to a health condition, the interviewer responds with a set of semi-structured questions. Recorded conversations were transcribed by trained medical transcribers according to a protocol established in conjunction with researchers [22].

### Participants and language samples

Interviewees were considered for inclusion in our AD group only if they met the following criteria: native English speaker; dementia specified as AD; dates of birth and conversation available; sex- and age-matched control available. These criteria resulted in 16 potential participants with AD. For each, the earliest dated intelligible audio-recorded conversation was extracted, edited, and coded according to CHAT guidelines [24]. This process resulted in speech samples that varied in length from 224 to 1,899 words. Sajjadi et al. [25], noting uncertainty around the word count necessary for a connected speech sample to realistically reflect language production, found 150- and 600-word transcripts comparable for analyses of connected speech by pwAD. However, longer texts are preferred when using lexical diversity to draw conclusions on a speaker's vocabulary [26]. As such, four transcripts with fewer than 500 words were excluded from analyses. The twelve remaining transcripts (Table 1) were cut to the end of the utterance containing the participant's 560[th] word, matching the length of the shortest transcript.

**Table 1. Group matching.**

| Measure | Group comparison | | |
|---|---|---|---|
| | pwAD (*n* = 12) | Controls (*n* = 12) | *t* |
| | Group mean (*SD*) | Group mean (*SD*) | (*p*) |
| | Range | Range | |
| Sex | 10 f, 2m | 10 f, 2m | |
| Age | 82.2 (7.1) | 80.6 (9.2) | 0.47 |
| | 68, 94 | 71, 101 | (*0.64*) |
| Overall tokens | 564.4 (5.53) | 562.8 (2.76) | 0.89 |
| | 560, 579 | 560, 569 | (*0.39*) |

Interviewees were considered for inclusion in our control group only if they met the following criteria: native English speaker; not reported to have AD or other neurological or psychiatric condition; dates of birth and conversation available. A control group was selected to match the group of pwAD in number, sex, and age. Groups were not matched for education because the CCC provides this only in broad ranges that do not facilitate meaningful matching—e.g., 1–8 years, 9–12 years, etc. Conversations for the control group were extracted, edited, coded, and shortened using the above process.

## Language analysis

Transcripts were analysed using CLAN, software designed to analyse transcripts conforming to CHAT guidelines [24]. Initial POS tags generated in CLAN were reviewed and inaccuracies that would affect analyses were corrected—for example, *hog* tagged as a verb when in context it had been used as a noun.

H1-H3 are on noun, verb, and pronoun quantities and N/V ratios, lexical diversity overall and for nouns and verbs, and copula use. Noun, verb, and pronoun quantities are measured via token counts. CLAN's default noun token counts were used in analyses. They include gerunds, ~*ing* verb forms functioning contextually as nouns and tagged as such in CLAN. Noun token counts do not include pronouns. For consistency with CLAN's default N/V ratio calculations, verb token counts include lexical verbs, copulas, and participles. Inclusion of copulas and participles in these counts requires a modification to CLAN code. CLAN's default pronoun token counts and N/V ratios were used in analyses. Lexical diversity of overall speech samples, of nouns, and of verbs are measured via type-token ratios (TTRs). TTR comparisons may be affected by differences in sample size—samples with more words may have lower TTRs due to increased likelihood of word re-use [27]. Speech samples in this study are matched for overall token count. However, they vary naturally in noun and verb counts. CLAN offers two alternative lexical diversity measures: moving average type-token ratio [26] and VocD [28]. Due to limitations in CLAN functionality for deriving these measures by POS, in this study, noun and verb diversity are measured via TTRs. For consistency and because overall token counts are matched, overall lexical diversity is also measured via TTRs. Overall TTRs consider all words in a speech sample. Noun TTRs consider all nouns and verb TTRs consider all verbs included in token counts. All TTRs used here are lemma-based [29], so that, e.g., *brother* and *brothers* are counted as two occurrences of the same word rather than as different words. Generation of lemma-based TTRs requires a modification to CLAN code. Copula production is measured via copula counts and ratios. CLAN's default copula, lexical verb, and participle counts were used to calculate copula-to-verb ratios as copula / (copula + lexical verb + participle).

H4 and H5 are on noun and verb frequency and AoA. Wordlists were generated in CLAN and used to obtain frequency and AoA measures from external databases. Wordlists included all tokens appearing in token counts except for gerunds, which were excluded due to discrepancies between the word form's POS in speech samples and the lemma's POS in databases. Word frequencies were obtained from WebCelex, the web-based interface to the CELEX lexical database [30]. Frequencies are reported as lemma appearances per million words. Regarding AoA data, few datasets account for distinctions between multiple POSs of a given word form, for example *walk* used as a verb or a noun. Viably scaled datasets that do this tend to be small [31]. This study reports group-level statistics based on mean AoAs extracted from the 30,121-word dataset of Kuperman et al. [32], which does not distinguish word forms by POS. Kuperman et al. [32] asked participants to estimate AoA as the age at which the participant believed they had learned the word. Ratings range from 1.5 to 25 years old and are rounded to two decimals. This study also reports advanced statistics based on ratings from the 2,694-word dataset of Bird et al. [33], the largest viably scaled AoA dataset to account for POS [cf. 31; see also Section 4.1.2]. Bird et al. [33] asked participants to estimate AoA on a 7-point Likert scale where each point corresponded to a period of two years including the age at which the participant believed they had learned the word (e.g., a rating of one indicating an age between 0 and 2 years old), with a rating of seven for any age over 13. Those authors then multiplied mean ratings by 100, so that final ratings are between 100 for a low AoA and 700 for a high AoA.

## Statistical analysis

The statistical software environment R [34] was used for all statistical analyses. H1-H3 were addressed using two-tailed independent samples *t*-tests that compared group mean noun, verb, and pronoun token counts, N/V ratios, TTRs overall and for nouns and verbs, and copula counts and ratios. Data used in these analyses met test assumptions. Alphas for *t*-tests were not adjusted for multiple comparisons because data analysed resulted from observations of natural phenomena [35]. Effect sizes are reported in Cohen's *d* [36]. Due to effects of sample length on TTR (see Section 2.3), reporting on TTR comparisons considers results of token count comparisons. H4 and H5 were addressed using linear mixed effects models to test predictors of frequency and AoA of nouns and verbs used. Model structures were determined *a priori* based on predictors of interest and included fixed effects of Group, POS, and Age. The continuous variable Age was centred but not scaled [37]. Sex was not included as a fixed effect because there were only two males in each group. Participant and Word were included in models as nested random effects, with frequency values or AoA ratings for a given word appearing as many times as the word appeared in a participant's speech sample. Because plots of preliminary models did not meet statistical assumptions, dependent variables were converted to natural logarithms for analysis. Group-level comparisons of overall, noun, and verb frequencies and AoAs are provided for descriptive purposes.

## Results

### POS production

Measures of POS production are presented in Table 2. pwAD produced significantly fewer noun tokens than controls ($p < 0.01$). This result was associated with a very large effect size ($d = 1.31$). Group differences in pronoun use did not reach significance ($p = 0.08$). However, a strong inverse relationship was present between noun and pronoun production by pwAD ($r = -0.66$, $p = 0.02$). The groups produced similar verb token counts and N/V ratios (both $p > 0.05$). All nonsignificant comparisons of POS production were associated with medium effect sizes.

**Table 2. POS production.**

| Measure | Group comparison | | | |
|---|---|---|---|---|
| | pwAD ($n$ = 12) | Controls ($n$ = 12) | $t$ | $d$ |
| | Group mean ($SD$) | Group mean ($SD$) | ($p$) | |
| | Range | Range | | |
| Noun tokens | 48.9 (10.5) | 65.3 (14.2) | -3.22 | 1.31 |
| | 35–72 | 41–89 | ($< 0.01$)** | |
| Verb tokens | 95.1 (7.9) | 100.3 (11.8) | -1.28 | 0.52 |
| | 81–103 | 75–119 | ($0.22$) | |
| Pronoun tokens | 119.2 (17.6) | 106.4 (16.5) | 1.83 | 0.75 |
| | 87–153 | 67–129 | ($0.08$) | |
| N/V ratio | 0.63 (0.21) | 0.79 (0.27) | -1.64 | 0.67 |
| | 0.40–1.11 | 0.46–1.39 | ($0.12$) | |

* $p < 0.05$

** $p \leq 0.01$

## Lexical diversity

TTRs of pwAD and controls are presented in Table 3. Overall TTRs were significantly lower for pwAD ($p$ = 0.02), indicating that they used a narrower range of words across all parts of speech than controls. This result was associated with a large effect size ($d$ = 1.02). Given the expected inverse relationship between token count and TTR, production of significantly fewer nouns by pwAD (see Sections 2.3 and 3.1) should be accompanied by higher noun TTRs than controls. However, group differences in noun TTRs were not significant ($p$ = 0.47), and pwAD produced lower mean noun TTRs than controls. Group differences in verb TTRs were not significant. Both nonsignificant TTR comparisons were associated with small effect sizes.

## Copula production

Two copula production measures were analysed—copula counts and copula-to-verb ratios. pwAD produced an average of 23.9 copulas ($SD$ = 7.8, range = 12–38) compared with controls' average of 22.6 ($SD$ = 5.8, range = 10–32). Group differences were not significant for this

**Table 3. TTRs.**

| Measure | Group comparison | | | |
|---|---|---|---|---|
| | pwAD ($n$ = 12) | Controls ($n$ = 12) | $t$ | $d$ |
| | Group mean ($SD$) | Group mean ($SD$) | ($p$) | |
| | Range | Range | | |
| Overall TTR | 0.29 (0.03) | 0.32 (0.03) | -2.5 | 1.02 |
| | 0.24–0.33 | 0.28–0.39 | ($0.02$)* | |
| Noun TTR | 0.64 (0.08) | 0.66 (0.08) | -0.74 [a] | 0.3 |
| | 0.54–0.80 | 0.53–0.78 | ($0.47$) | |
| Verb TTR | 0.39 (0.06) | 0.40 (0.05) | -0.61 | 0.25 |
| | 0.33–0.51 | 0.34–0.49 | ($0.55$) | |

* $p < 0.05$

** $p \leq 0.01$

[a] Interpretation of this result should consider group differences in noun token production (see text).

measure ($t$ = 0.48, $p$ = 0.64, $d$ = 0.19). pwAD produced an average copula-to-verb ratio of 0.25 ($SD$ = 0.08, range = 0.15–0.4) compared with controls' average of 0.23 ($SD$ = 0.07, range = 0.1–0.31). Group differences were not significant for this measure ($t$ = 0.75, $p$ = 0.46, $d$ = 0.31). These nonsignificant comparisons were associated with small effect sizes.

## Noun and verb frequencies

Group-level statistics on the frequencies of words produced are presented in Table 4. While pwAD exhibited a tendency to produce words of higher frequencies than controls, two-tailed independent samples $t$-tests indicated that group means did not differ significantly for overall words, nouns, or verbs ($p > 0.05$).

Frequency data were available for 3,664 of 3,716 nouns and verbs spoken by participants. These included 562 of 587 nouns and 1,139 of 1,141 verbs spoken by pwAD (mean frequencies per million words: nouns 354 ± 484, verbs 11365 ± 15700) and 761 of 784 nouns and 1,202 of 1,204 verbs spoken by controls (nouns 310 ± 456, verbs 10635 ± 15175). Within both groups, frequencies were higher and variances greater for verbs than nouns (Fig 1). Qualitative inspection of the data indicated that several verbs, most notably *be* (38301 appearances per million words), were several times more frequent than both other verbs (next most frequent: *have*, 13494) and the most frequent nouns used within either group (pwAD: *one*, 2073; controls: *time*, 1971).

Frequency values were converted to natural logarithms for analysis due to heteroscedasticity in the residuals of a preliminary model that used actual values. As seen in Table 5, a linear mixed effects model using log frequencies revealed a significant main effect of POS ($p < 0.01$). Main effects of Group and Age were not significant. Significant two-way interactions were present between Group and POS ($p < 0.01$) and Group and Age ($p < 0.03$). The two-way interaction between POS and Age did not reach significance ($p = 0.07$). The three-way interaction between Group, POS, and Age was significant ($p < 0.01$).

Model predictions are presented visually in Fig 2. Changes with age in frequencies of nouns used are apparent in both groups. Noun frequencies rise with age for controls, indicating use of less advanced nouns. By contrast, noun frequencies decrease with age for pwAD, suggesting use of more advanced nouns. Verb frequencies remain relatively stable with age in both groups, increasing slightly for pwAD, who use verbs of higher frequencies than controls. Participants in both groups use verbs of higher frequencies than their own nouns.

**Table 4. Overview of word frequencies.**

| Measure | Group comparison | | |
|---|---|---|---|
| | pwAD ($n$ = 12) | Controls ($n$ = 12) | $t$ |
| | Group mean ($SD$) | Group mean ($SD$) | ($p$) |
| | Range | Range | |
| Overall frequency | 8956 (1069) | 8661 (771) | 0.77 |
| | 7188–11008 | 7192–9968 | (0.45) |
| Noun frequency | 355[a] (148) | 328[a] (160) | 0.44 |
| | 199–712 | 140–764 | (0.66) |
| Verb frequency | 11400[a] (3029) | 10793[a] (2869) | 0.5 |
| | 6920–17066 | 5427–15014 | (0.62) |

* $p < 0.05$, ** $p \le 0.01$

[a] Frequency means reported in the text differ from those reported in Table 4. Means reported in the text consider all nouns or verbs produced within a group, while means reported in Table 4 are group means based on participant means.

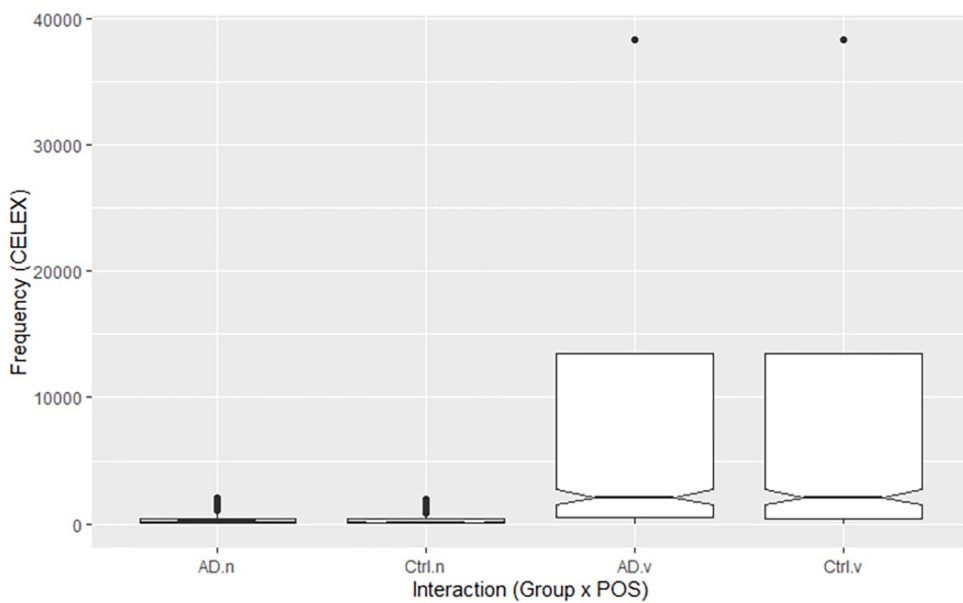

**Fig 1. Frequency by POS and group.** *Note.* AD = pwAD, Ctrl = controls, n = nouns, v = verbs.

Follow-up testing was conducted to explore the unexpected finding that pwAD used nouns of lower log frequencies with age, while controls used nouns of higher log frequencies. Of specific interest was how these trends might relate to noun token counts, as well as how noun token production might relate to age. As seen in Fig 3, changes with age in noun token production were not significant. Production decreased moderately for controls while increasing moderately for pwAD.

## Noun and verb AoAs

Prior to AoA analyses, a Pearson's correlation was conducted to assess the relationship between frequency and AoA of all words used in this study. This relationship, while significant, was weak ($r = -0.11$, $p < 0.01$).

Group-level statistics on the AoAs of words produced are presented in Table 6. On average, pwAD produced nouns and verbs of higher mean AoAs—i.e., more advanced nouns and verbs

**Table 5. Frequency results.**

| Predictor/interaction | Estimate | Standard error | t | p |
|---|---|---|---|---|
| Intercept | 5.19 | 0.12 | 42.73 | $< 0.01$** |
| Group | 0.03 | 0.16 | 0.16 | 0.87 |
| POS | 1.46 | 0.11 | 13.66 | $< 0.01$** |
| Age (centred) | -0.02 | 0.02 | -1.22 | 0.24 |
| Group x POS | -0.38 | 0.13 | -2.84 | $< 0.01$** |
| Group x Age (centred) | 0.05 | 0.02 | 2.36 | 0.03* |
| POS x Age (centred) | 0.03 | 0.02 | 1.8 | 0.07 |
| Group x POS x Age (centred) | -0.06 | 0.02 | -3.05 | $< 0.01$** |

* $p < 0.05$

** $p \leq 0.01$

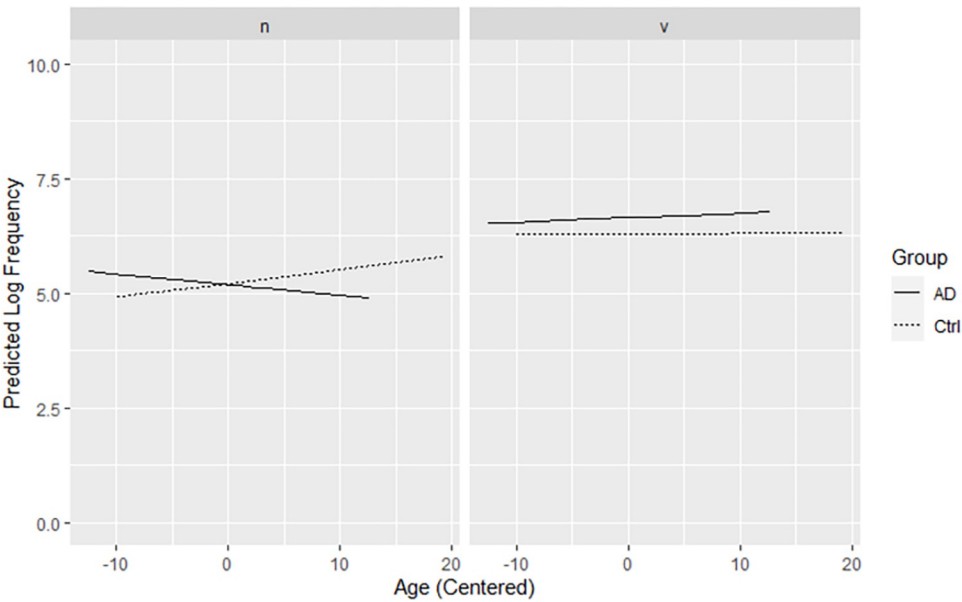

**Fig 2. Predicted effects of group, age, and POS on log frequency of words used.** (n) For controls, noun frequencies rise with age, indicating use of less advanced nouns. For pwAD, noun frequencies decrease with age, suggesting use of more advanced nouns. (v) Verb frequencies remain relatively stable with age in both groups, rising slightly for pwAD, who use verbs of higher frequencies than controls. (n and v) Participants in both groups use verbs of higher frequencies than their own nouns. *Note.* AD = pwAD, Ctrl = controls, n = nouns, v = verbs.

—than controls. However, two-tailed independent samples *t*-tests indicated that group means did not differ significantly for overall words, nouns, or verbs ($p > 0.05$).

AoA data were available for 2,722 of 3,716 nouns and verbs spoken by participants [33]. These included 276 of 587 nouns and 1,037 of 1,141 verbs spoken by pwAD (mean AoA: nouns 273 ± 72, verbs 280 ± 43) and 329 of 784 nouns and 1,080 of 1,204 verbs spoken by

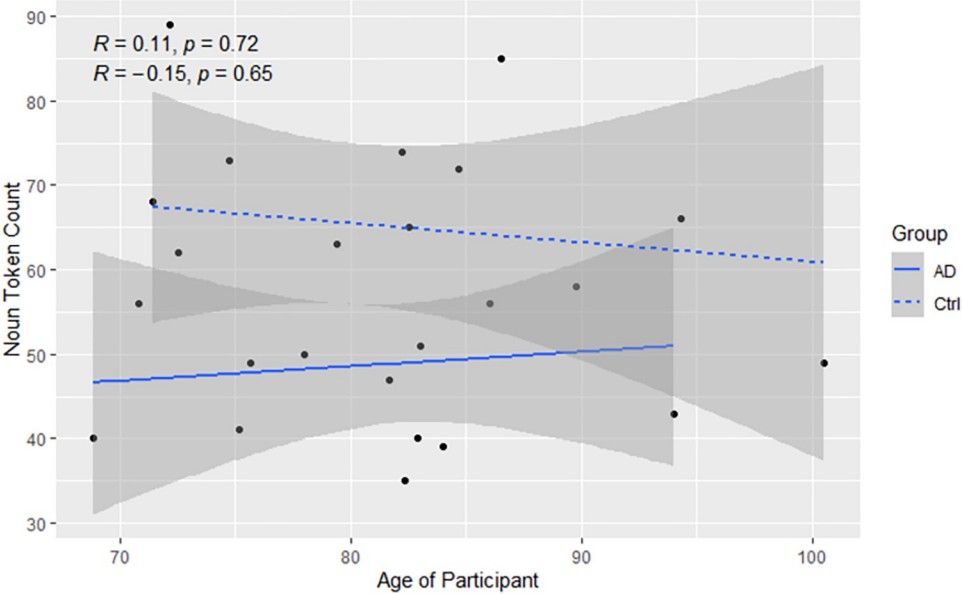

**Fig 3. Relationship between age and noun token production, by group.**

**Table 6. Overview of AoAs.**

| Measure[a] | Group comparison | | |
|---|---|---|---|
| | pwAD (*n* = 12) | Controls (*n* = 12) | *t* |
| | Group mean (*SD*) | Group mean (*SD*) | (*p*) |
| | Range | Range | |
| Overall AoA | 4.48 (0.11) | 4.51 (0.15) | -0.65 |
| | 4.3–4.7 | 4.3–4.9 | (*0.52*) |
| Noun AoA | 275.5[b] (19.1) | 267.6[b] (14.4) | 1.15 |
| | 245–308 | 250–292 | (*0.26*) |
| Verb AoA | 279.6[b] (8.6) | 276.6[b] (8.6) | 0.87 |
| | 266–295 | 265–298 | (*0.4*) |

\* $p < 0.05$, \*\* $p \leq 0.01$

[a] Overall AoAs come from the ratings of Kuperman et al. [32]. Noun and verb AoAs come from the ratings of Bird et al. [33]. See Section 2.3 for further information on these ratings datasets.

[b] AoA means reported in the text differ from those reported in Table 6. Means reported in the text consider all nouns or verbs produced within a group, while means reported in Table 6 are group means based on participant means.

controls (nouns 268 ± 73, verbs 276 ± 43). AoAs were similar between groups for the respective word classes (Fig 4), with variances greater for nouns than verbs.

AoA ratings were converted to natural logarithms for analysis due to a positive skew in the residuals of a preliminary model that used actual ratings. As seen in Table 7, a linear mixed effects model using log AoAs revealed significant main effects of Group ($p < 0.01$) and POS ($p = 0.01$). The interaction between these terms was also significant ($p < 0.01$). The main effect of Age was not significant, and no interactions including the Age term were significant.

Model predictions are presented visually in Fig 5. Between groups, pwAD use nouns of higher AoA than controls, while the groups differ little in AoA of verbs used. Within groups, pwAD use nouns of higher AoA than their own verbs, while controls use verbs of higher AoA than their own nouns.

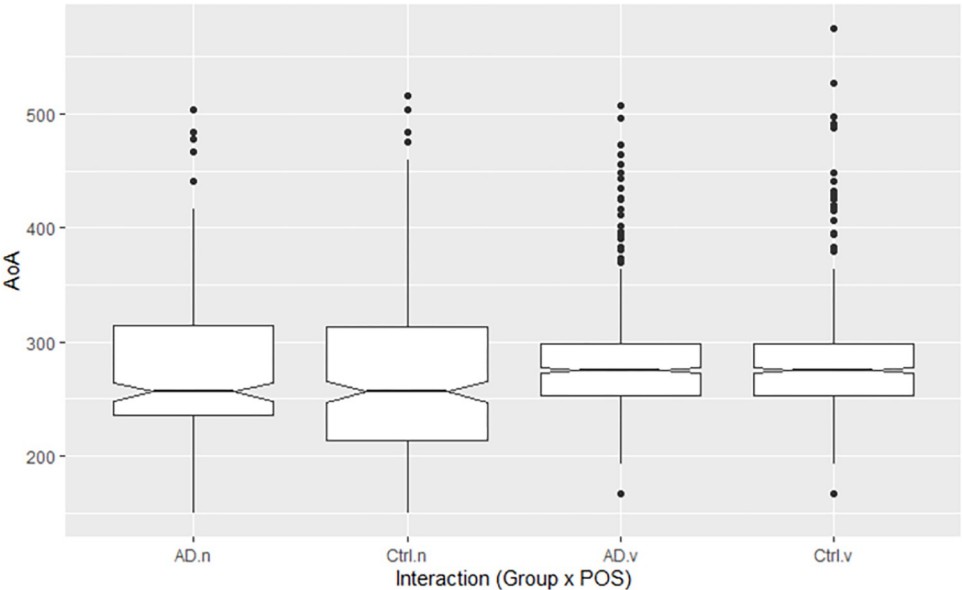

**Fig 4. AoA by POS and group.** *Note.* AD = pwAD, Ctrl = controls, n = nouns, v = verbs.

**Table 7. AoA results.**

| Predictor/interaction | Estimate | Standard error | t | p |
|---|---|---|---|---|
| Intercept | 5.64 | 0.01 | 406.4 | < 0.01** |
| Group | -0.10 | 0.02 | -6.06 | < 0.01** |
| POS | -0.03 | 0.01 | -2.48 | 0.01** |
| Age (centred) | 0.00 | 0.00 | 0.75 | 0.46 |
| Group x POS | 0.13 | 0.02 | 8.27 | < 0.01** |
| Group x Age (centred) | -0.00 | 0.00 | -0.81 | 0.43 |
| POS x Age (centred) | -0.00 | 0.00 | -1.14 | 0.26 |
| Group x POS x Age (centred) | 0.00 | 0.00 | 0.18 | 0.86 |

* $p < 0.05$

** $p \leq 0.01$

Follow-up testing was conducted to explore the unexpected finding that pwAD used nouns of higher log AoAs than controls. Of specific interest was how this finding might relate to group differences in noun token counts. As such, relationships were examined between participants' noun token counts and mean log AoAs of nouns used. As seen in Fig 6, increased use of nouns was not significantly associated with decreases in log AoA of nouns ($p = 0.08$).

## Discussion

This study aimed to investigate whether pwAD differed from healthy age-matched controls in their use of words of different POS in conversational interviews. Speech samples of 12 pwAD and 12 controls were matched for overall word count and analysed for noun, verb, and pronoun quantities, N/V ratios, overall and noun and verb TTRs, copula counts and ratios, and noun and verb frequencies and AoAs. pwAD produced significantly fewer nouns than

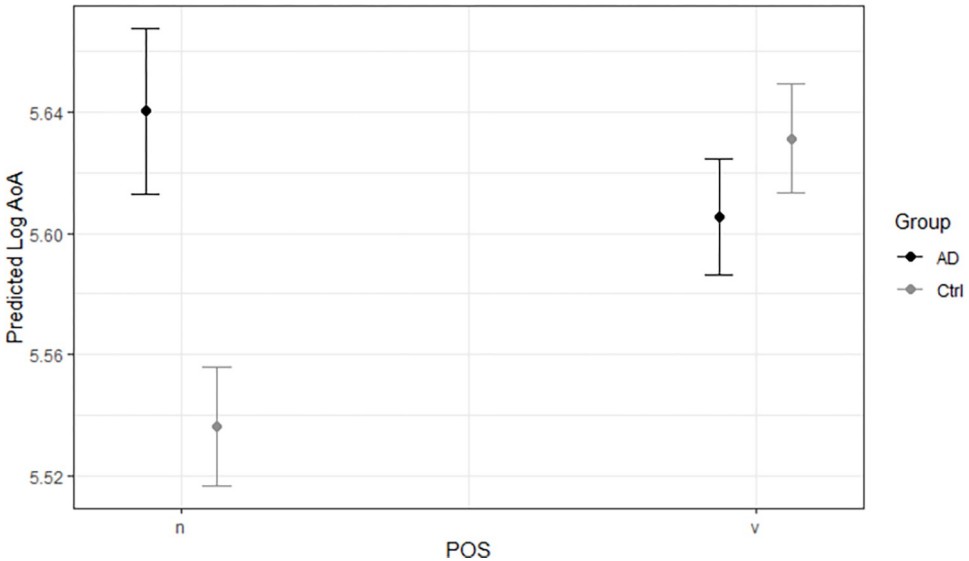

**Fig 5. Predicted effects of group and POS on log AoA of words used.** (Left side) pwAD use nouns of higher AoA than controls. (Right side) The groups differ little in AoA of verbs used. (Both sides) pwAD use nouns of higher AoA than their own verbs, while controls use verbs of higher AoA than their own nouns. *Note*. AD = pwAD, Ctrl = controls, n = nouns, v = verbs.

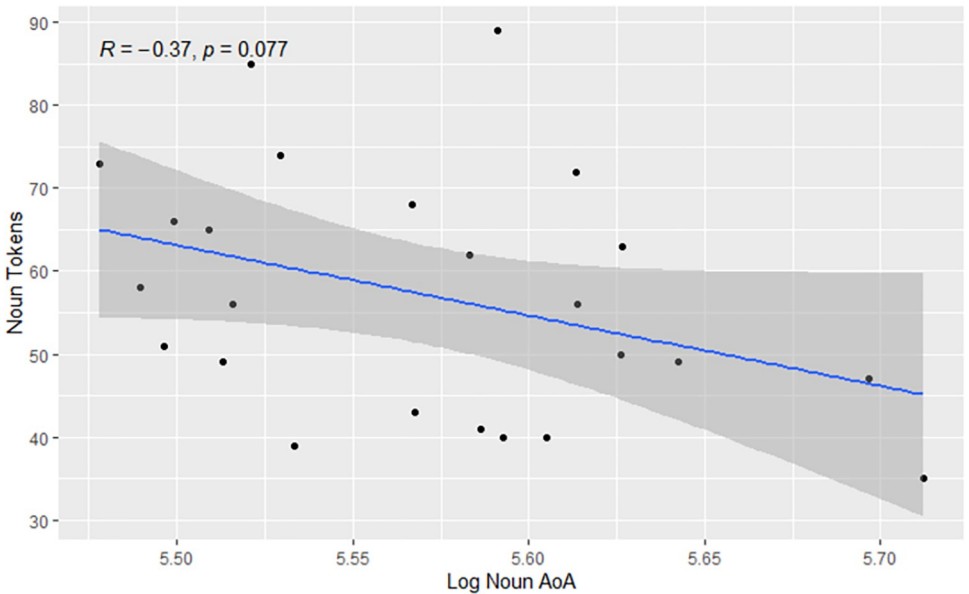

**Fig 6. Relationship between noun use and log noun AoA.**

controls. They also produced lower overall TTRs, while noun TTR comparisons were not significant despite the group differences in noun production. Differences in pronoun production did not reach significance ($p = 0.08$). The groups did not significantly differ in verb token counts, N/V ratios, verb TTRs, or copula use. Frequency was predicted by a main effect of POS, two-way interactions between group and POS and group and age, and a three-way interaction between group, POS, and age. AoA was predicted by main effects of group and POS and a two-way interaction between group and POS.

Findings on nouns and verbs and their implications for research on language in ageing and AD are discussed in Section 4.1. Implications of these findings for communicative interventions are discussed in Section 4.2. Study limitations are discussed in Section 4.3.

## Noun and verb use by pwAD and controls

**H1-H3: Word production and lexical diversity.** The finding here of decreased overall lexical diversity in speech by pwAD, indicating use of a narrower range of words, is consistent with consensus findings [7, 9, 17]. Evidence here indicates that this overall reduction is linked to changes in noun use. While direct comparison of noun TTRs suggested group differences were not significant, this result is likely attributable to differences in quantities of nouns produced, as TTRs are inversely related to token counts [27]. pwAD used significantly fewer nouns than controls, which should have resulted in higher mean noun TTRs if the diversity among nouns used by the groups was similar. Instead, noun TTRs of pwAD were lower than those of controls despite their lower token counts, suggesting decreased diversity among nouns produced by pwAD. These findings of decreases in noun production and diversity are consistent with prior findings on noun use in discourse by pwAD [9, 15]. Use of fewer nouns and of a decreased range of words overall, specifically of nouns, by pwAD are likely related through a process by which retrieval issues lead to the replacement of nouns, a wide-ranging, open word class, with a necessarily limited number of words from the closed class pronouns [16]. The inverse relationship here between noun and pronoun use by pwAD is evidence of this replacement process. Its effects on overall lexical diversity are suggested in the

combination of decreases in overall TTR and noun token counts alongside nonsignificant increases in pronoun use compared to controls. Relationships between changes in noun and pronoun use and decreased overall lexical diversity in AD are also suggested in prior reporting of similar combinations of findings [9, 16, 17]. Further studies should explore the role of retrieval issues in these changes.

In contrast to the differences for nouns, pwAD here did not differ from controls in verb token production, verb TTRs, or copula use. These findings contradict prior indications that in discourse, pwAD produce higher proportions of verbs, particularly generic verbs such as copulas, and a less diverse range of verbs than cognitively healthy controls [9, 15, 18]. Despite group differences in noun production, N/V ratios did not differ by group, likely relating to a nonsignificant decline in verb production by pwAD. N/V ratios can be used to identify a retrieval issue for one of these two major word classes relative to the other [e.g., 38]. However, the implication of this N/V ratio comparison is that no such issue is present for these pwAD. This contrasts with the findings on nouns and pronouns, which appear to suggest noun retrieval difficulties. Direct comparisons of word use by POS—either quantities in standardised samples, as used here, or proportions or percentages of all words—likely provide the better indication of POS-specific retrieval in discourse.

**H4 and H5: Noun and verb frequencies and AoAs.**   It was expected that pwAD would use words of higher frequencies and lower AoAs than controls. Lower word frequency and higher word AoA lead to difficulties for both the cognitively healthy and pwAD on discrete language tasks [13, 39]. This may relate to the role of these learning history variables in declarative memory and semantic connectedness [7, 12]. Past findings have suggested that this difficulty with more advanced words may translate to reliance by pwAD on simple, generic alternatives in discourse [9, 16, 18]. These findings are frequently derived from simple comparisons of group means [e.g., 16, 18]. Here, preliminary *t*-testing indicated no group differences in frequencies or AoAs of overall words, nouns, or verbs produced. Results of a linear mixed effects model also indicated little overall difference in frequencies of nouns or verbs used by group; however, age affected noun frequencies differently within each group. Frequencies of nouns used by pwAD decreased with age, suggesting use of more advanced nouns, while frequencies of nouns used by controls rose with age. A separate linear mixed effects model found that pwAD produced nouns of higher AoA than controls regardless of age, seemingly indicating consistent use of more advanced nouns. These differences in findings by statistical method highlight the benefits of using more robust tests. Use of mixed modelling here provided information on effects of not only age but also properties of individual words and their use within and across participants. Discussion below of psycholinguistic properties of words produced will focus on results of these models.

With age, pwAD produced nouns of lower frequencies and exhibited nonsignificant increases in noun production. This combination of findings appears to suggest improving noun production with age in pwAD. Unfortunately, the CCC lacks demographic information relevant to interpretation of these findings. Participant education levels were only available in broad ranges, so that we were unable to meaningfully explore contributions of education to language production by pwAD. It is possible that older pwAD here were more educated than younger ones, contributing to use of nouns of lower frequencies [40]. Alternatively, the group's age range (68–94) and the ages of the youngest pwAD (68, 70) at time of conversation suggest the presence within the group of both late-onset AD and the rarer early-onset form (EOAD) [41]. EOAD is associated with more rapid cognitive decline and different language symptoms [42]. It is possible that EOAD in younger pwAD contributed to these findings of age effects. However, date or age at diagnosis—information that would have been useful in determining the presence of EOAD [41, 43]—were unavailable. These findings highlight the need for future

studies to examine effects of education on noun production in conversation by pwAD and to compare noun production by people with early- versus late-onset AD. The vagueness around interpretation of these findings reinforces the importance of accounting for relevant demographic variables during study design, recruitment, and data analysis.

In the present study, cognitively healthy adults over age 70 used nouns of higher frequencies with age in spontaneous speech. A nonsignificant decrease in noun use with age was also present in this group. This combination of findings suggests possible changes in noun use in discourse production in advanced ageing. Healthy older adults are known to experience word retrieval difficulties on single-word tasks. However, it has been suggested that vocabulary growth with age compensates for retrieval difficulties on single-word and discourse tasks [44]. Kavé et al. [45] found that despite increasing noun retrieval difficulty on naming and fluency tasks, healthy speakers aged 20 to 85 produced less frequent nouns with age in picture descriptions. The authors attributed this to older speakers' larger vocabularies. The present findings are based on production of less constrained discourse by participants older on average than the oldest 30 participants in that study. These findings necessitate larger studies of changes to noun use not across the lifespan, but specifically in advanced ageing. They also underscore the usefulness of detailed reporting on performance by controls, which is at times lacking in studies attributing language changes to AD [7]. Our follow-up frequency analysis considered trends in noun use in both groups rather than exclusively probing the unexpected production of nouns of lower frequencies with age by pwAD. This revealed potential changes in healthy ageing that provided perspective on the apparent changes in pwAD. Future studies attributing language changes to AD should ensure they provide similar context by either including middle-aged comparison groups or analysing changes with age in healthy controls.

A follow-up analysis suggests the unexpected finding that pwAD used nouns of higher AoA than controls should not be taken at face value, as this finding may relate to significant group differences in noun token production. An inverse relationship ($p = 0.08$) was present in these data between quantities of noun tokens produced by participants and mean log AoAs of those nouns. Positive skews were also present in AoAs both overall and for nouns prior to conversion for statistical analyses, suggesting heavy reliance on nouns of earlier AoAs. Such distributions are in line with Zipf's [46] observation that speakers generally prefer less advanced words. While Zipf's observation was based on word frequencies, it also applies to AoA and, by way of these variables, to semantic connectivity [12, 47, 48]. Thus, production of more noun tokens would likely lead to higher proportions of nouns acquired at earlier ages, potentially explaining the finding here that controls used nouns of lower AoA than pwAD. Low numbers of noun tokens in AoA analyses may have also factored into this finding. Fewer than half the nouns appearing for each group in frequency analyses appeared in AoA analyses (frequency vs. AoA nouns: pwAD 562 vs. 276; controls 761 vs. 329). This likely related to our use of the Bird et al. [33] AoA dataset. While this may be the largest set of viably scaled AoA ratings to account for a word form's POS (see below), the inclusion of only 2,694 words in that dataset is a limitation that must be considered in interpreting the present findings. Another relevant consideration is the lack of information on participant education—pwAD may have been more educated than controls, and this might have contributed to production of more advanced nouns.

Results of linear mixed effects models and *t*-tests suggest that verb frequencies and AoAs differed little by group, with effects of age on frequency less pronounced for verbs than nouns. Together, then, findings from H1-H5 suggest unimpaired verb production in spontaneous speech by pwAD while providing mixed evidence of noun production deficits. These differences in findings by POS are underscored by significant main effects of POS in both frequency and AoA models. Past findings on discrete word production also suggest POS effects in

performance by pwAD [7]. However, in those tasks, pwAD have greater difficulty with verbs than nouns. This combination of findings suggests that an advantage for nouns on single-word tasks does not translate to discourse. Findings that pwAD are no more impaired in describing actions than people or objects in pictures [4, 49] may be seen as further evidence of this. Poorer retrieval of verbs than nouns in single-word tasks has been attributed to weaker semantic organizations among verbs [50, 51]. Discourse contexts may mitigate verb retrieval deficits through the necessary interactions between verbs and words around them. These interactions recruit syntactic and morphological processes, which are relatively preserved in pwAD [7]. Task effects arising from differences in processing demands would likely relate further to the engagement of distinct neural areas depending on whether language is produced or comprehended in context or in isolation [52]. Potential effects of task type on retrieval of nouns and verbs by pwAD would be best investigated via within-participant comparisons of performance on single-word and discourse tasks. Analyses of functional imaging would provide further information on neural correlates of language processing and effects on performance resulting from changes to specific neural areas in AD.

Findings here on verbs are based on higher and more similar token counts than those on nouns, demonstrating that in addition to processing demands, verbs and nouns also differ in the extent to which they appear in discourse. Speakers' heavier reliance on verbs, together with the larger number of nouns than verbs in the English lexicon, contribute to much higher mean frequencies for verbs than nouns. In the present data, mean frequencies of verbs were 32 times higher than those of nouns (Table 4). Despite this, verbs are acquired later in life [14]. POS differences in frequency and AoA have ramifications for interpretations of word use and lexical-semantic decline in AD. While we did not find group differences in overall word frequency, multiple discourse studies have reported higher overall frequencies for pwAD than controls [19]. An example is Kavé and Dassa [16], who elicited picture descriptions by pwAD nearly twice as long as those of controls. Their groups did not differ in percentages of words that were verbs, suggesting more verbs appeared for pwAD in analyses. pwAD also relied more on pronouns and less on nouns. Thus the finding of higher frequencies among words used by pwAD may simply reflect group differences in numbers of nouns, verbs, and pronouns produced. POS differences also apply to overall AoA comparisons. These were not significant here despite higher noun AoA for pwAD than controls. Yeung et al. [20], though, found pwAD to use words of higher AoA than controls in Cookie Theft picture descriptions. Because this task commonly elicits words of lower AoA, the authors interpreted the finding as a sign that pwAD were more likely to make off-topic remarks. However, the authors also reference rate of noun and verb phrases as an explanatory factor in word-finding difficulties for pwAD. While this is not elaborated on, group differences in noun and/or verb production may have contributed to the overall AoA finding. Accounting for this potential confound in studies of psycholinguistic effects in language production by pwAD would allow for more conclusive testing of whether pwAD replace advanced words of a given POS with less specific ones.

Frequency and AoA differences between nouns and verbs are also relevant to the design of single-word studies. Stimuli in those studies are often matched for frequency and AoA, including at times across POS. Druks et al. [50], for example, found both pwAD and controls to name objects faster and more accurately than actions. Object and action stimuli were matched for AoA, with frequencies only marginally higher for verbs. Since verbs tend to have much higher frequencies than nouns, a reduction in these differences may have resulted in use of more advanced verbs, factoring into relative success with object stimuli. A better practice may be to match stimuli according to word class norms, as in White-Devine et al. [51], who found pwAD but not controls more accurate in naming objects.

More broadly, differences highlighted here between nouns and verbs—in frequency, AoA, proportion of words spoken, and context-dependent processing demands—are suggestive of deeper differences between the word classes. Those differences should be accounted for as potential confounds in design stages of investigations of word-class specific impairments in pwAD. Conclusions should not be drawn based strictly on within-group comparisons of performance across word classes. POS-specific impairments in pwAD should be framed within the context of any differences, or lack thereof, by word class for controls.

Differences were also present here across psycholinguistic variables. pwAD used nouns of similar frequencies but higher AoAs than controls. Age factored into predictions of noun frequencies but not AoAs. Nouns were acquired earlier than verbs despite being less frequent in general usage. Frequency and AoA have been claimed to reflect the same information [53], a claim supported by tendencies toward earlier acquisition of words of higher frequencies. However, the relationship between frequencies and AoAs of words used in this study was weak, and conclusions that could be drawn on noun use differ by measure. Sailor et al. [54] also found differences in these measures, reporting that words produced on a semantic fluency task were of lower frequencies but lower AoAs than those produced on a letter fluency task.

While these findings seem to indicate that the measures are not proxies for one another, they may result from differences in data quality. Frequency values, including those used here, are generally based on counts of a lemma's appearances in a large corpus. Lemmas correspond to meanings rather than word forms, thereby providing more specific information to corpus users. AoA measures frequently involve ratings based on adults' perceptions of their own or their child(ren)'s learning. While these methods allow for collection of large datasets, flaws related to subjectivity are inherent [see 55]. More objective data collection methods such as recording or testing children engender separate issues, including difficulty obtaining large amounts of data. Regardless of collection method, AoA data often do not distinguish between meanings of a word form, including multiple POS [31]. This is problematic considering differences in AoA by POS. Brysbaert and Biemiller [31] constructed a comprehensive set of acquisition norms for 44,000 word meanings, including multiple meanings of a word form, from which POS can be inferred. Construction of this dataset involved retrospective changes to data from the testing of US children beginning in grade 4, including the creation of a grade 2 norm. The authors provide a formula by which to derive an AoA from a word's grade level of acquisition. While these norms attempt to address flaws in prior AoA datasets, the authors acknowledge crudeness in their scale, based as it is on grade, rather than age, of acquisition. The scale is divided into increments of two grades, or years, beginning at US grade level 2, or age seven. It therefore essentially provides a default AoA of seven for any word acquired by this age, at which children generally already possess a large vocabulary. The higher end of their scale is also problematic in that testing of adults was limited to students enrolled in thirteen to sixteenth years of schooling, with words acquired after year 13 automatically assigned a grade level of 14. Among other potential uses, a large, objective, precisely scaled set of AoA norms that accounts for multiple word meanings would facilitate improved understanding of relationships between frequency and word acquisition.

## Implications for communicative interventions

The use by pwAD of nearly twice as many verbs as nouns, together with potentially facilitative effects of context on verb production, necessitate research on communicative interventions targeting verbs. Such interventions have shown success in post-stroke aphasia rehabilitation. Improvements experienced by people with aphasia following verb treatments include greater generalization across word, sentence, and discourse levels than is seen with noun treatments.

Improvements appear to be independent of both the underlying verb deficit and the nature of the therapy, though they may be restricted by co-occurring syntactic deficits [56]. Aphasia researchers have recommended not treating verbs in isolation but instead focusing on their argument structures and associated nouns [57]. Such an approach may play to relative strengths of pwAD. As a progressive condition, AD should be considered less conducive to rehabilitative treatments than post-stroke aphasia. However, prior studies on verb acquisition in AD [58, 59] suggest some potential for targeting verbs in language maintenance or restoration. These studies found pwAD able to acquire grammatical features of new verbs. They were less successful with argument structures. This finding, however, may have related to aspects of research methodology—the researchers viewed argument structure as a semantic property of a verb, contradicting the consensus view that it is a syntactic property [7]. Given that syntax is relatively spared in pwAD, a syntax-based approach to acquisition of argument structure may facilitate production of both verbs and nouns in discourse. Study is needed of whether verb treatments for aphasia may, with or without modification, improve communication by pwAD.

Decreased noun production by pwAD highlights the need to address noun use in communicative interventions, since these decreases have broad implications for pwAD and their caregivers. Nouns are content words that convey meaningful detail, specifying for example entities that perform or receive an action. Their replacement with less specific pronouns likely factors into perceptions of spoken discourse by pwAD as vague or empty [9]. The replacement of open-class with closed-class words also results in decreased lexical diversity. Adults have been shown to judge children whose speech exhibits low lexical diversity as less appealing, mature, or talkative than those with higher lexical diversity [60]. A question for future research is whether pwAD are judged negatively based on decreased lexical diversity. Such judgments might lead caregivers to engage less in conversation with pwAD, contributing to social isolation and declines in mental health and quality of life for both parties [1, 2]. Mental state and cognitive activity, including social interaction, can affect cognitive abilities [61, 62], so that this process may speed declines experienced by pwAD. Word retrieval in context and caregiver responses to simplified language production are therefore potential intervention targets.

## Limitations

This study has limitations of which the reader should be aware. Severity of cognitive decline in pwAD could only be considered at a general level. While a uniform statement is available on AD stages of CCC participants, information is not provided for individuals. Education level, a predictor of both word use and decline in AD [40, 63], is provided in broad ranges only and so was not considered here. Due to small group sizes and specifically a lack of male participants, analyses were not controlled for gender, another predictor of decline in AD [64]. Larger studies of the lexical-semantic measures investigated here should account for progression of cognitive decline, education, and gender. Speech samples were included based on a word count threshold and most were artificially shortened. While this practice facilitated statistical analyses, it ignored potential contributors to variation among speech samples, which may include cognitive ability. These data are not controlled for interviewer or conversation topic. While this has the benefit of more closely simulating real-life communication, these factors can influence the communicative participation and word choice of interviewees. Limitations related to existing AoA datasets are discussed in Section 4.1.2. Despite its limitations, this study has provided information on lexical-semantic changes that may accompany ageing and AD while highlighting methodological considerations for improved investigation of these changes.

## Conclusion

This study compared conversational use of words of different POS by pwAD and healthy age-matched controls. Previous findings have suggested that breakdowns in semantic representations in AD lead to frequent reuse of generic words, including pronouns, copulas, and other high-frequency nouns and verbs. In this study, pwAD produced fewer nouns and more pronouns than controls, leading to decreased lexical diversity. Frequencies of nouns produced did not differ by group, though age affected these differently. pwAD produced nouns of lower frequencies with age, possibly due to differences in education or in age of AD onset. Controls produced nouns of higher frequencies with age, which may suggest difficulty retrieving nouns in conversation in advanced ageing. pwAD used nouns of higher AoA than controls, a finding that may relate to group differences in noun quantities. Use of verbs differed little by group. Overall, aside from increased pronoun use, pwAD did not tend to produce less sophisticated words than controls. Across groups, POS significantly affected word frequency and AoA, with verbs having much higher frequency values than nouns. These findings highlight the importance of breaking down findings by POS when assessing language use. Future reporting on lexical diversity, frequency, and AoA in speech by pwAD should account for POS effects. These findings also provide rationale for targeting both nouns and verbs in communicative interventions.

## Acknowledgments

The Medical University of South Carolina (MUSC) and the University of North Carolina at Charlotte (UNCC) are thanked for providing access to Carolina Conversations Collection data. Professor Boyd Davis of UNCC and senior systems engineer Paul Arrington of MUSC are thanked for facilitating its use. University of Canterbury associate professor Daniel Gerhard and senior lecturer Donald Derrick are thanked for their advice on statistical procedures. Professor Sarah Wallace and Dr. Tamiko Azuma are thanked for helpful comments on a previous version of this manuscript.

## Author Contributions

**Conceptualization:** Eric Williams, Catherine Theys, Megan McAuliffe.

**Data curation:** Eric Williams.

**Formal analysis:** Eric Williams, Catherine Theys, Megan McAuliffe.

**Investigation:** Eric Williams.

**Methodology:** Eric Williams, Catherine Theys, Megan McAuliffe.

**Software:** Eric Williams.

**Visualization:** Eric Williams.

**Writing – original draft:** Eric Williams.

**Writing – review & editing:** Catherine Theys, Megan McAuliffe.

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
