## [Decision Letter · Decision Letter 0]

25 Apr 2023

PONE-D-23-03295Lexical-semantic properties of verbs and nouns used in conversation by people with Alzheimer's diseasePLOS ONE

Dear Dr. Williams,

Thank you for submitting your manuscript to PLOS ONE. After careful consideration, we feel that it has merit but does not fully meet PLOS ONE’s publication criteria as it currently stands. Therefore, we invite you to submit a revised version of the manuscript that addresses the points raised during the review process. I hope that you find the reviewer's comments helpful. Additionally, I would recommend that you consider augmenting certain sections of your literature review to include the work of, for example, Boyd Davis, Paul Darden, et al.

We look forward to receiving your revised manuscript.

Kind regards,

Christopher James Hand, Ph.D., M.Sc., M.A., PgCAP

Academic Editor

PLOS ONE

Journal Requirements:

Reviewers' comments:

Reviewer's Responses to Questions

**Comments to the Author**

1. Is the manuscript technically sound, and do the data support the conclusions?

Reviewer #1: Yes

Reviewer #2: Yes

2. Has the statistical analysis been performed appropriately and rigorously? 

Reviewer #1: Yes

Reviewer #2: Yes

3. Have the authors made all data underlying the findings in their manuscript fully available?

Reviewer #1: Yes

Reviewer #2: No

4. Is the manuscript presented in an intelligible fashion and written in standard English?

Reviewer #1: Yes

Reviewer #2: Yes

5. Review Comments to the Author

Reviewer #1: Two small changes: the new standard term is /people living with dementia/ -- I don't think, from personal observation when collecting a lot of the data in the collection, that they ALL had Alzheimer's. Your abbreviation throughout should be PLWD

Set off the hypotheses as a bulleted list or it gets confusing

Reviewer #2: The authors aimed to investigate whether people with Alzheimer’s disease differed in their use of words from different grammatical categories compared to age matched controls. Data was from a US-based digital archive of interviews. Speech samples from 12 people with AD and 12 controls were matched for total word count. The AD group was found to produce less nouns than the control group, and type-token ratios were lower, indicating use of a narrower range of words across all word types, compared to controls. Analysis of the frequencies and age of acquisition of words used revealed that, with increasing age, the AD group used less frequent nouns, while the control group used nouns of higher frequency. The authors discuss education level and type of AD as potential reasons for the findings re the effect of age. They make the excellent point that there is a lack of studies of change in noun use in advanced ageing. The authors discuss the fact that their results indicate unimpaired verb use by the AD group, while research on single word naming reveals a verb disadvantage in AD. There is an excellent discussion of potential problems in matching nouns and verbs on age of acquisition and frequency in studies of single word naming.

This was a clearly written paper (in the main, please see below), with well-founded measures employed in thorough analyses of the data. Implications of the findings for intervention are clearly drawn at the end of the paper. I believe the study is definitely worthy of publication. I just had some minor points for consideration,

Line 74-75 ‘Interestingly, AoA comparisons have not indicated reliance on simpler words by pwAD’ – could the authors please explain what is meant here.

Line 140-141 ‘Groups were not matched on education because this was provided in 4-year ranges’ – could the authors please explain what is meant here.

6. PLOS authors have the option to publish the peer review history of their article (what does this mean?). If published, this will include your full peer review and any attached files.

Reviewer #1: No

Reviewer #2: No

---

## [Author Response · Author response to Decision Letter 0]

27 May 2023

This information is also copied and pasted in from the response to reviewers (E Williams):

A. Editor’s comments: 

I hope that you find the reviewer's comments helpful. Additionally, I would recommend that you consider augmenting certain sections of your literature review to include the work of, for example, Boyd Davis, Paul Darden, et al.

We found the reviewers’ comments extremely helpful.

We also appreciated your suggestions of references to help establish the scientific context for our work. We identified two studies by Boyd Davis and colleagues that presented ramifications of changes to language and communication in Alzheimer’s disease. These now appear in the text and reference list as references 2 and 3. Use of these resources has resulted in changes to lines 39-40 (unmarked version) of the Introduction. Reference number 2 also supports and allows for expansion on a point made in the Discussion section; as a result, changes were also made to lines 583-584.

B. Reviewer 1’s comments

the new standard term is /people living with dementia/ -- I don't think, from personal observation when collecting a lot of the data in the collection, that they ALL had Alzheimer's. Your abbreviation throughout should be PLWD

We acknowledge Reviewer 1’s familiarity with the CCC. We are happy to make this change if necessary, pending confirmation of whether the reviewer’s comment considers our description of inclusion criteria in the “Participants and language samples” section. The CCC includes Alzheimer’s disease and dementia as two distinct selection criteria under the Condition heading. We specifically included only data from participants meeting the Alzheimer’s criterion—no one who was only described using the broader term dementia. Our description thus reads, “Interviewees were considered for inclusion in our AD group only if they met the following criteria: … dementia specified as AD…” (bold letters added for emphasis). Can the reviewer please clarify whether this comment applies to participants designated specifically in the CCC search filters as having AD? If it does, we are happy to apply this change. If the comment no longer applies, but the reviewer feels our description could be clarified, we are happy to take suggestions on how to do so.

Set off the hypotheses as a bulleted list or it gets confusing

We have made this change and agree that it improves readability. Please see lines 95 to 104 of the unmarked version.

C. Reviewer 2’s Comments

Line 74-75 'Interestingly, AoA comparisons have not indicated reliance on simpler words by pwAD' - could the authors please explain what is meant here.

We have revised and added detail to this statement. It now reads, “Interestingly, findings from picture description tasks have not indicated that pwAD rely on words of lower AoA [8, 19]. Fraser et al. [8] included AoA in their investigation of language features that might help distinguish pwAD from healthy older people but did not find it to contribute to identification of pwAD. Yeung et al. [19], meanwhile, actually found pwAD to use words of higher AoA than controls, a finding they attributed to a tendency by pwAD to produce utterances that were inappropriate to the task.” In the current unmarked version, that information can be found in lines 68 to 73.

Line 140-141 'Groups were not matched on education because this was provided in 4-year ranges' - could the authors please explain what is meant here.

We thank the reviewer for highlighting this. In following up on the comment, we found our initial description not to be fully accurate and we have rephrased as: “Groups were not matched for education because the CCC provides this only in broad ranges that do not facilitate meaningful matching—e.g., 1-8 years, 9-12 years, etc.” As a result of the revision process, this information now appears in lines 141-143. We have also revised wording now in line 411, as that sentence reiterates this point on education.

---

## [Decision Letter · Decision Letter 1]

29 Jun 2023

Lexical-semantic properties of verbs and nouns used in conversation by people with Alzheimer's disease

PONE-D-23-03295R1

Dear Dr. Williams,

We’re pleased to inform you that your manuscript has been judged scientifically suitable for publication and will be formally accepted for publication once it meets all outstanding technical requirements.

Kind regards,

Christopher James Hand, Ph.D., M.Sc., M.A., PgCAP

Academic Editor

PLOS ONE

Additional Editor Comments (optional):

I am pleased to report that two reviewers have now endorsed publication of your manuscript.

They have provided a little additional feedback; please consider this when preparing the final version of your manuscript for resubmission.

Please also carefully check the manuscript and full references for any typographic errors, formatting inconsistencies, etc.

Reviewers' comments:

Reviewer's Responses to Questions

**Comments to the Author**

1. If the authors have adequately addressed your comments raised in a previous round of review and you feel that this manuscript is now acceptable for publication, you may indicate that here to bypass the “Comments to the Author” section, enter your conflict of interest statement in the “Confidential to Editor” section, and submit your "Accept" recommendation.

Reviewer #1: All comments have been addressed

Reviewer #2: All comments have been addressed

2. Is the manuscript technically sound, and do the data support the conclusions?

Reviewer #1: Partly

Reviewer #2: Yes

3. Has the statistical analysis been performed appropriately and rigorously? 

Reviewer #1: Yes

Reviewer #2: Yes

4. Have the authors made all data underlying the findings in their manuscript fully available?

Reviewer #1: Yes

Reviewer #2: No

5. Is the manuscript presented in an intelligible fashion and written in standard English?

Reviewer #1: Yes

Reviewer #2: Yes

6. Review Comments to the Author

Reviewer #1: I still think that AOA is not needed although I think you gave a very good discussion of its limitations. That's why I said 'partly' above.

Reviewer #2: Thank you for clearly responding to the two queries.

All required questions have been answered and all responses meet formatting specifications.

7. PLOS authors have the option to publish the peer review history of their article (what does this mean?). If published, this will include your full peer review and any attached files.

Reviewer #1: No

Reviewer #2: **Yes: **Jackie Masterson

---

## [Editor Report · Acceptance letter]

24 Jul 2023

PONE-D-23-03295R1 

Lexical-semantic properties of verbs and nouns used in conversation by people with Alzheimer's disease 

Dear Dr. Williams:

I'm pleased to inform you that your manuscript has been deemed suitable for publication in PLOS ONE. Congratulations! Your manuscript is now with our production department. 

Kind regards, 

on behalf of

Dr. Christopher James Hand 

Academic Editor

PLOS ONE